# Melatonin Improves Glucose Homeostasis and Insulin Sensitivity by Mitigating Inflammation and Activating AMPK Signaling in a Mouse Model of Sleep Fragmentation

**DOI:** 10.3390/cells13060470

**Published:** 2024-03-07

**Authors:** Seok Hyun Hong, Da-Been Lee, Dae-Wui Yoon, Jinkwan Kim

**Affiliations:** 1Sleep Medicine Institute, Jungwon University, Goesan-gun 28204, Chungcheongbuk-do, Republic of Korea; mlmljk3620@gmail.com (S.H.H.); dabin050912@korea.ac.kr (D.-B.L.); 2Department of Biomedical Laboratory Science, Jungwon University, Goesan-gun 28204, Chungcheongbuk-do, Republic of Korea; 3Department of Health and Safety Convergence Science, Graduate School, Korea University, Seoul 02841, Republic of Korea

**Keywords:** sleep fragmentation, metabolic dysfunction, AMP-activated protein kinase (AMPK), melatonin

## Abstract

Sleep fragmentation (SF) can increase inflammation and production of reactive oxygen species (ROS), leading to metabolic dysfunction. SF is associated with inflammation of adipose tissue and insulin resistance. Several studies have suggested that melatonin may have beneficial metabolic effects due to activating AMP-activated protein kinase (AMPK). However, it is unclear whether melatonin affects the AMPK signaling pathway in SF-induced metabolic dysfunction. Therefore, we hypothesize that SF induces metabolic impairment and inflammation in white adipose tissue (WAT), as well as altered intracellular homeostasis. We further hypothesize that these conditions could be improved by melatonin treatment. We conducted an experiment using adult male C57BL/6 mice, which were divided into three groups: control, SF, and SF with melatonin treatment (SF+Mel). The SF mice were housed in SF chambers, while the SF+Mel mice received daily oral melatonin. After 12 weeks, glucose tolerance tests, insulin tolerance tests, adipose tissue inflammation tests, and AMPK assessments were performed. The SF mice showed increased weight gain, impaired glucose regulation, inflammation, and decreased AMPK in WAT compared to the controls. Melatonin significantly improved these outcomes by mitigating SF-induced metabolic dysfunction, inflammation, and AMPK downregulation in adipose tissue. The therapeutic efficacy of melatonin against cardiometabolic impairments in SF may be due to its ability to restore adipose tissue homeostatic pathways.

## 1. Introduction

Sleep fragmentation (SF), which is characterized by frequent arousals and interruptions of sleep continuity, is becoming more common due to environmental factors and various sleep disorders such as obstructive sleep apnea (OSA) and insomnia [1]. Over the past decade, extensive research has demonstrated significant links between chronic SF, whether induced experimentally or in patient populations, and markedly increased risks of obesity, impaired glucose tolerance, insulin resistance, type 2 diabetes, hypertension (HTN), and other features of metabolic syndrome [2]. Although the exact underlying mechanism between SF and metabolic dysfunction is not clearly elucidated, the activation of oxidative stress and increased inflammation have been proposed to play an important role in these associations [3,4,5]. 

Melatonin is a hormone secreted by the pineal gland that regulates the sleep–wake cycle and has been found to have a wide range of metabolic regulatory bioactivities. Accumulating data from preclinical and experimental studies also support that melatonin may have beneficial effects on insulin signaling, adipocyte differentiation, and lipid metabolism in obesity, as well as inflammation in diabetes and related comorbidities [6,7,8,9].

AMP-activated protein kinase (AMPK), a key cellular energy sensing complex [10,11], has emerged as a mediator between the misalignment of circadian rhythms and metabolic homeostasis [12,13]. Activation of AMPK has been associated with beneficial effects on metabolism, such as improving insulin sensitivity and ameliorating obesity [14,15]. Interestingly, melatonin has been reported to activate AMPK, providing a potential mechanistic link between melatonin and improved cardiometabolic outcomes [16,17]. Recently, several studies have also shown that SF induces insulin resistance by promoting inflammation in adipose tissue through nicotinamide adenine dinucleotide phosphate (NADPH) oxidase [18]. Thus, the present study aims to investigate the potential role and underlying mechanisms of melatonin in ameliorating metabolic dysfunction induced by SF, with a particular focus on AMPK activation in white adipose tissue (WAT), in a mouse model. 

## 2. Materials and Methods

### 2.1. Schematic Diagram and Graphical Summary of a Mouse Model of SF

A schematic diagram of the experiment and a graphical summary of the mouse model of SF are summarized in Figure 1.

### 2.2. Experiment Design and Animals

Adult male C57BL/6 mice (8 weeks old, 22–23 g; Eumseong, Chungcheongbuk-do, Republic of Korea) were obtained from the DBL laboratory for the study. Mice were maintained on a normal chow diet and housed in a controlled environment with a regular 12 h light–dark cycle at a constant temperature (24 ± 2 °C) and ad libitum access to food and water. Mice were randomly divided into three groups: control mice (*n* = 10), mice housed in standard housing conditions and sleep fragmented (SF, *n* = 10), and melatonin-treated SF mice (SF+Mel, *n* = 10) housed in an SF chamber as described previously (model 80391; Lafayette Instrument, Lafayette, IN, USA) [19]. Due to limitations on blood volume and adipose tissue size, we used separate sets of mice to study glycemic response and reactive oxygen species (ROS) detection on adipose tissue stromal vascular fraction (SVF). Body weight was measured weekly for each mouse. Mice were sacrificed after 12 weeks of exposure to SF (Figure 1A), and tissue samples were collected for further analysis. Animal experiments were performed according to a protocol approved by the IACUC of Jungwon University (JWU-IACUC-2022-3).

### 2.3. Sleep Fragmentation and Melatonin Treatment

The SF group and SF+Mel group mice were exposed to SF for 12 weeks. The machine used to induce sleep fragmentation has previously been described [19,20]. In brief, the timer of the moving sweep bar was set for mice with severe sleep apnea, waking more than 30 times per hour. The sweep bar was set to move once every 2 min and moved when the light was on (9:00 a.m. to 9:00 p.m.). SF+Mel mice were orally administered melatonin (Sigma, St. Louis, MO, USA) at a concentration of 40 mg/kg daily [8]. 

### 2.4. Biochemical Analysis

A glucose tolerance test (GTT) and insulin tolerance test (ITT) were performed on all three groups at 4 weeks and 12 weeks of SF. The animals were fasted for 6 h, with water available ad libitum. An oral GTT was conducted using sterile glucose (2 g/kg body weight with oral zondae). For the insulin tolerance test, the mice were intraperitoneally injected with insulin (St Lenexa, KS, USA) at a concentration of 0.75 units/kg body weight. Blood samples for GTT and ITT were collected from the tail vein of each mouse at 0, 30, 60, 90, and 120 min. Blood glucose levels were measured using a glucometer (Barojan, Handok, Seoul, Republic of Korea). Additionally, the area under the curve (AUC) was calculated for GTT and ITT using the trapezoidal method. Triglyceride (TG) and total cholesterol levels were also assessed using an Accutrend Plus system (Roche, Basel, Switzerland) [21]. Serum C-peptide levels were measured using an enzyme immunoassay kit (Novus Biological, E Easter Ave, Centennial, CO, USA) according to the manufacturer’s protocol. The linear range of the C-peptide was 0.16–10 ng/mL, with both intra- and inter-individual coefficients of variation of up to 6.0%.

### 2.5. Immunofluorescent Staining for Detecting the AMPK and F4/80 in WAT 

After mice were sacrificed, epidymal WAT samples were fixed with 4% paraformaldehyde for 24 h. The samples were transferred to 15% sucrose (JUNSEI Co., Ltd., Chuo-ku, Tokyo, Japan) for 12 h and 30% sucrose for 24 h at 4 °C. The WAT samples were embedded in O.C.T. compound [22]. Sections of 12 μm were obtained using a cryomicrotome. The cryosections were stained with a fluorescein isothiocyanate (FITC)-conjugated primary antibody against AMPK alpha 1 (Novus Biologicals, E Easter Ave, Centennial, CO, USA) and F4/80 (Invitrogen, Carlsbad, CA, USA) at a dilution of 1:50 overnight at 4 °C. After washing with phosphate-buffered saline (PBS), the sections were costained with 1 uL Hoechst 33342 (Thermo Fisher, Waltham, MA, USA) for 15 min at room temperature. Fluorescence images were obtained by excitation at 610 nm and collected at 600–700 nm using a confocal laser scanning microscope (LSM 800, Carl Zeiss, Jena, Germany). Image analysis was performed using ZEISS ZEN lite 3.8 (Oberkochen, Germany). 

### 2.6. Quantitative Real Time Polymerase Chain Reaction

To analyze the gene expression of AMPK and inflammatory signaling pathway-related phenotypes, we extracted total RNA from the WAT using a RNeasy Lipid Tissue Mini kit (Qiagen, Hilden, Germany), following the manufacturer’s protocol. We then synthesized cDNA using a Tetro cDNA synthesis kit (Meridian Bioscience Inc., Cincinnati, OH, USA) and conducted quantitative real-time polymerase chain reaction (qRT-PCR) tests using a StepOne PlusTM real-time PCR system (Applied Biosystems, Waltham, MA, USA). The gene expression assay employed commercially available specific TaqMan primers and probes for *Prkaa1* (assay ID: Mm01296700_m1), *Stk11* (assay ID: Mm00488470_m1), *Camkk2* (assay ID: Mm00520236_m1), *IL1β* (assay ID: Mm00434228_m1), *IL6* (assay ID: Mm00446190_m1), *TNF* (assay ID: Mm00443258_m1), *IL10* (assay ID: interleukin 10), *TGF-β1* (assay ID: Mm01178820_m1), *ADGRE1* (assay ID: Mm00802529_m1), *CCL2* (assay ID: Mm00441242_m1), and *GAPDH* (assay ID: Mm99999915_g1). All reactions were performed in triplicate. The ^2^(−ΔΔCT) method was utilized to compare gene expression among various groups [23]. *GAPDH* was used as an internal control.

### 2.7. Flow Cytometry Analysis and Western Blotting

To detect AMPK and ROS levels in SVF derived from WAT, flow cytometry analysis was performed. SVF was isolated by incubating and digesting fat cells from WAT with type I collagenase (Gibco, New York, NY, USA) and type II dispase (Sigma, St. Louis, MO, USA) in Dulbecco Modified Eagle Medium (Thermofisher, Waltham, MA, USA) for 25 min at 37 °C. The cell suspensions were filtered through a 40 μm strainer (SPL, Republic of Korea) and then centrifuged at 1500 rpm for 5 min to separate the floating adipocytes from the SVF pellet. To detect mitochondrial ROS in SVF, we used the MitoSOX flow cytometry assay kit (Invitrogen, Carlsbad, CA, USA) according to the manufacturer’s protocol. This assay selectively detects superoxide in the mitochondria of living cells. A total of 1.5 × 10^6^ cells obtained from suspended SVF pellet were stained with 5 μM Mitosox-red and incubated for 40 min at 37 °C [18,24]. In addition, intracellular staining was also performed to examine AMPK levels in SVF. The SVF pellets were fixed with Fix/Perm buffer at room temperature and washed with Perm/wash (Biolegend, San Diego, CA, USA). The SVF pellet was then resuspended with staining buffer (Biolegend, San Diego, CA, USA), and 3 × 10^6^ cells were stained with FITC-conjugated AMPK alpha1 antibody (NOVUSBIO, E Easter Ave, USA) for 30 min in the dark. Flow cytometric analyses were conducted using a BD Accuri C6 flow cytometer (BD Biosciences, San Jose, CA, USA), and the obtained data were analyzed using Flow JO version 10 software (Tree Star, San Carlos, CA, USA). To examine the phosphorylation level of AMPK at threonine 172 (pAMPK), protein was extracted from WAT using protein extraction solution (Intron-Bio Ltd., Ggyeonggi-do, Republic of Korea). The protein samples (20 ug) were separated by electrophoresis on 10% SDS-PAGE and transferred onto a PVDF membrane. After blocking with 5% BSA in TBST, the membranes were incubated with primary antibodies p-AMPK (Cell signaling, 1:2000), AMPK alpha (Cell signaling, 1:4000), and β-actin-conjugated HRP (Santa Cruz, 1:1000) at 4 °C overnight. The following day, the membrane’s pAMPK was incubated with HRP-conjugated anti-rabbit IgG (Cell Signaling, 1:5000) at room temperature for 1 h. Proteins were visualized using a chemiluminescent peroxidase substrate (Amersham), and the blots were detected using the ChemiDoc system (Bio-Rad, Hercules, CA, USA). The signals were quantified using ImageJ ver. 1.54 analysis software.

### 2.8. Statistical Analysis

The data are presented as the mean ± standard error (SE). Statistical analyses were performed using Mann–Whitney U-tests or Kruskal–Wallis tests to examine the difference between groups, unless otherwise stated in the figure legends. All statistical analyses were performed using SPSS software (version 25.0, IBM Corp., Armonk, NY, USA). Statistical significance was identified at the 0.05 significance level.

## 3. Results

### 3.1. The Change in Body Weight after Exposure to SF and Treatment with Melatonin in Mice

Figure 2 shows the changes in body weight in three groups of mice after 12 weeks of exposure to SF. The SF group exhibited a significant increase in body weight starting at 4 weeks of exposure (Figure 2A). At the end of the 12-week period, both weight and weight gain in the SF group were significantly higher than in the control group (*p* < 0.05). In addition, the SF+Mel group exhibited a significant decrease in body weight and weight gain compared to the SF group after 12 weeks of melatonin treatment (*p* < 0.05). 

### 3.2. The Alteration of Lipid Profiles after Exposure to SF and Treatment with Melatonin

To compare the lipid profiles of the control, SF, and SF+Mel groups at 4, 8, and 12 weeks, we measured the levels of total cholesterol and TG in the blood samples. There were no significant differences in total cholesterol or TG levels among the three groups at 4 or 8 weeks (*p* > 0.05). However, after at 12 weeks, the TG level in the SF group was significantly higher than that in the control group (Figure 3B). After 12 weeks of melatonin treatment, the SF+Mel group showed a significant decrease in TG levels compared to the SF group (Figure 3B).

### 3.3. Melatonin Improved Glycemic Dysregulation in SF Exposed Mice

GTT and ITT were performed at 4 and 12 weeks in three groups. The SF mice exhibited significantly higher glycemic levels in both GTT and ITT compared to the control mice at 12 weeks (Figure 4B,C), but this was not observed after 4 weeks of exposure in SF mice. Interestingly, the glucose levels obtained from GTT and ITT in the SF+Mel groups were significantly reduced compared to those of SF mice after 12 weeks of melatonin treatment (Figure 4B,C). The glucose AUC also exhibited significant decreases (Figure 4D,E). Additionally, we measured C-peptide levels, known to be an important marker of β-cell function and insulin production [25,26], to examine the differences between the three groups. As a result, the C-peptide levels in the SF mouse group were significantly lower than those in the control mice (SF mice vs. control mice, 0.31 ± 0.2 ng/mL vs. 0.55 ± 0.2 ng/mL, *p* < 0.05). In contrast, the levels were increased in the SF+Mel group after 12 weeks of melatonin treatment. The SF mice had a level of 0.31 ± 0.2 ng/mL, while the SF+Mel mice had a level of 0.65 ± 0.2 ng/mL (*p* < 0.05, Figure 4F).

### 3.4. SF Exhibited Increased ROS and Inflammation in WAT, and These Improved with Melatonin Treatment for 12 Weeks

To investigate whether SF could increase ROS and inflammation in WAT, we measured ROS levels using a commercially available Mito-Sox kit (Invitrogen, USA). As shown in Figure 5, SF mice exhibited a significant increase in ROS levels in the SVF of WAT compared to control mice (control vs. SF, 12.6% vs. 31.0%). Additionally, we administered melatonin for 12 weeks to examine its effect on ROS levels in SF mice. The study revealed that SF+Mel mice had a significantly lower level of ROS in SVF compared to SF mice (Figure 5A, SF vs. SF+Mel, 31.0% vs. 14.2%). Additionally, Figure 5B displays a representative image of F4/80, which is not only a marker for inflammatory macrophages, but also plays a crucial role in regulating adipose tissue function and insulin sensitivity. After 12 weeks of treatment with melatonin, the F4/80 levels in WAT were significantly decreased in the SF+Mel group compared to the SF group. To support these phenomena in WAT, we performed gene expression assays using the specific primers mentioned above. When comparing gene expression among the three groups, we observed an increase in the pro-inflammatory cytokines *IL-1β* and *TNF-α* in the SF group. Additionally, *F4/80* and *MCP-1*, which are widely used as markers of macrophage inflammation, also significantly increased. However, treatment with melatonin alleviated the gene expression of these cytokines and macrophages. 

### 3.5. SF Decreased AMPK Level and Melatonin Treatment Improved AMPK Levels in WAT for 12 Weeks

To investigate whether SF could decrease the AMPK levels in WAT, the AMPK levels were measured using an immunofluorescent staining assay and flow cytometry with a specific antibody. The results showed that the AMPK levels in SVF derived from the WAT of SF mice decreased compared to that of control mice, but it was increased in SF+Mel mice after 12 weeks of melatonin treatment (Figure 6A, control vs. SF vs. SF+Mel, 23.3% vs. 13.8% vs. 20.3%). The confirmation of this phenomenon was achieved using an immunofluorescent technique (Figure 6B). Similar findings were observed when comparing gene expression among the three groups. Additionally, we performed Western blotting using a specific antibody to detect phosphorylation of AMPK at threonine 172 (pAMPK), which plays an important role in AMPK activity [27]. After 12 weeks of melatonin treatment, pAMPK levels in the WAT of SF+Mel mice significantly increased compared to SF mice, which had significantly decreased levels.

### 3.6. The Gene Expression Levels of Glycolipid Pathway Associated with AMPK in WAT after Exposure to SF and Melatonin Treatment

We examined the gene expression levels of glycolipid metabolic pathways associated with AMPK after 12 weeks of exposure to SF and treatment with melatonin in WAT. We observed a significant increase in the expression of both *HMGCR* (3-hydroxy-3-methyl-glutaryl-CoA reductase) and *SREBP-1* (sterol regulatory element-binding protein-1) in SF mice. However, these significantly decreased after melatonin treatment (Figure 7, *p* < 0.05). In addition, the expression level of *GLUT4* (glucose transporter 4), which plays an important role in regulating glucose metabolism, significantly decreased in WAT of SF mice. However, it increased after melatonin treatment. 

## 4. Discussion

SF is a common component of various sleep disorders, such as OSA, insomnia, and periodic leg movement. It is characterized by repeated interruptions in sleep without necessarily leading to full awakenings. SF has become increasingly prevalent in modern society. Extensive evidence has revealed that it is linked not only to various cardiometabolic diseases, such as obesity, impaired glucose metabolism, insulin resistance, and HTN [28,29], but also to deleterious consequences for daytime sleepiness and cognitive function [30]. A significant finding in this study is that chronic SF in a mouse model resulted in metabolic impairment, regardless of any effects from sleep loss or disruption of circadian rhythms. Most strikingly, it reveals the potent effects of melatonin treatment in improving the dysfunction of glycolipid metabolism induced by SF. After 12 weeks of melatonin supplementation, SF mice exhibited marked improvements in weight, glycemic regulation, TG levels, and AMPK signaling pathways in adipose tissue. This mimics highly prevalent conditions such as OSA and insomnia, where arousals and SF represent the primary underlying abnormality [2,18,29]. The ability of SF alone to recapitulate the metabolic sequelae observed in sleep apnea patients further cements the crucial mechanistic role it plays in cardiometabolic risk. 

Melatonin is a hormone secreted by the pineal gland that regulates the sleep–wake cycle and has metabolic regulatory roles in the body. Accumulating studies have shown that melatonin plays an important role in the regulation of glucose homeostasis, lipid metabolism, oxidative stress, and inflammation, making it relevant in metabolic disease research [8,9,31]. We found that melatonin effectively mitigated the metabolic disturbances induced by 12 weeks of SF, demonstrating promise in treating obesity and insulin resistance. This suggests that melatonin could be a viable preventative or therapeutic agent for patients suffering from SF. However, it is crucial to elucidate the mechanisms underlying the metabolic benefits of melatonin treatment. Based on the results, we discuss the key findings associated with SF-induced metabolic dysfunction in a logical order. First, it was found that SF stimulates immune cell infiltration and inflammation in WAT, which is consistent with previous research [18]. After 12 weeks of SF, markers of tissue inflammation and macrophage accumulation were universally heightened. The study found that there was an increase in the gene expression of cytokines such as IL-1β and TNF-α, as well as F4/80 and MCP-1, in adipose tissue. There are established indicators of immune cell migration and pro-inflammatory polarization in adipose tissue [5,32,33,34]. Chronic inflammation in WAT is known to drive insulin resistance and metabolic deterioration through pathways such as JNK and NF-kB [33]. Therefore, the significant increase observed in this study likely constitutes a major cause of the metabolic dysfunction induced by SF. Second, the study showed a significant increase in the production of mitochondrial ROS in WAT after 12 weeks of SF. This is closely related to inflammation, as oxidative stress can trigger pro-inflammatory signaling cascades that disrupt insulin pathways [5,35]. Mitochondrial dysfunction is also inherent in the progression of obesity and diabetes [36,37]. Increased ROS generation is likely both a result and a cause of metabolic impairment induced by SF [5]. Third, melatonin administration significantly suppressed inflammation and oxidative stress caused by SF in WAT. This was evidenced by reduced immune cell infiltration, pro-inflammatory gene expression, and mitochondrial superoxide levels after 12 weeks of melatonin treatment. Previous studies have confirmed the well-documented anti-inflammatory and antioxidant properties of melatonin and suggested that these properties contribute to its metabolic benefits [38,39]. Therefore, preventing the activation of damaging inflammatory and redox pathways in WAT may have prevented the downstream insulin signaling deficits. This idea is supported by the observed improvements in glucose homeostasis [6,40]. Fourth, more importantly, we found that AMPK dysregulation played a significant role in SF-induced metabolic impairment, which was partially reversible with melatonin. AMPK is a protein kinase that monitors and regulates the energy state of a cell. This kinase is activated when cellular levels of adenosine triphosphate (ATP) decrease, indicating either high energy consumption or decreased energy supply [10,27]. In this study, we observed that SF significantly suppressed both AMPK levels and the phosphorylation level of AMPK at threonine 172 (pAMPK) in WAT, while melatonin increased the expression back towards normal function. Reduced AMPK expression could decrease fatty acid oxidation, glucose uptake, and mitochondrial capacity [41]. These factors are closely linked to insulin sensitivity and metabolic health [10,42]. Correspondingly, SF mice exhibited decreased gene expression of GLUT4, along with increased expression of lipogenic genes such as HMGCR and SREBP-1. The study suggests that melatonin may improve glucose metabolism by restoring energetic balance in WAT through the rescue of AMPK expression and related pathways [10,41]. The results suggest that melatonin treatment can activate AMPK phosphorylation at threonine 172, which may lead to functional activation [10,41]. Previous studies have shown that melatonin can activate and regulate glucose levels in metabolic tissues by supporting AMPK [17,41,43]. This is likely achieved by synchronizing hypothalamic circadian clocks, which in turn regulate peripheral AMPK rhythms through downstream neurohormonal networks [12,44]. Disruption of circadian control contributes to metabolic disorders, such as sleep apnea and shift work [5,37,45]. The chronobiotic properties of melatonin may restore the metabolic oscillations necessary for proper AMPK functioning [12,46]. Further studies should investigate the interplay between melatonin, AMPK signaling, and circadian rhythmicity to uncover valuable targets for preventing or managing cardiometabolic disease. Moreover, it is worth considering whether melatonin protects mitochondria by scavenging reactive oxygen species and maintaining cellular energy production in SF mice. This is because the activation of AMPK can vary in different types of tissues [43,47]. Thus, the metabolic benefits are likely due to a combination of reduced inflammation, a balanced redox state, and optimized AMPK signaling. Thus, future research should aim to distinguish the relative contributions of these interconnected pathways in detail.

It is important to discuss the limitations of our study. Firstly, we used a mouse model of SF, which does not fully replicate the complex physiological and metabolic disturbances observed in clinical conditions such as sleep apnea. Therefore, it is difficult to extrapolate the magnitude of effects to actual patients in clinical settings. Secondly, although the mice showed weight gain and metabolic changes, the degree of impairment appeared relatively mild to moderate over 12 weeks. High-risk patients may have different treatment responses due to more pronounced or progressive metabolic disease. Therefore, longer SF exposures, similar to those seen chronically in humans, could increase clinical relevance. Additionally, the study convincingly demonstrates the therapeutic efficacy of melatonin; however, the underlying mechanisms were only partially characterized. Several pathways, including inflammation, oxidative stress, and AMPK signaling, have significant crosstalk. More dedicated molecular approaches are necessary in order to elucidate the proportional contribution of each process to metabolic protection. Additionally, the generalizability of the effects is unclear. To establish broader applicability to patients, efficacy should be tested in older mice, females, other strains, or disease models (e.g., diet-induced obesity).

Despite the aforementioned drawbacks, this study has several advantages. Firstly, it reveals melatonin as a promising therapeutic agent for ameliorating the metabolic disturbances induced by chronic SF. This is a unique study exploring the effect of melatonin in a model that closely mimics sleep apnea or environmentally induced SF. The study results are compelling, as melatonin supplementation prevented weight gain, insulin resistance, hyperlipidemia, and adipose tissue inflammation triggered by SF. Additionally, the study provides important mechanistic insight into pathways underlying the pathogenesis of SF-induced metabolic dysfunction, including dysfunction of adipose tissue AMPK signaling. The activation of AMPK may be a key mediator of the beneficial metabolic actions of melatonin following chronic SF. These insights enhance our understanding of how SF enhances cardiometabolic risk and reveal potential therapeutic targets. The study utilizes an array of techniques, from gene expression to histology, to comprehensively characterize metabolic, inflammatory, and intracellular signaling responses to SF and melatonin.

## 5. Conclusions

In summary, this study provides insight into the pathogenesis of SF-induced metabolic impairment, while uncovering melatonin as a promising therapeutic agent in this setting. The study indicates that chronic SF triggers inflammation, oxidative stress, and AMPK downregulation, changes that are largely mitigated by long-term melatonin supplementation. The restoration of WAT function appears to be central to the efficacy of melatonin in improving systemic metabolism. These findings should encourage research into clinical applications for high-risk patients with SF, such as OSA and insomnia. Larger trials are needed to confirm whether melatonin therapy can reduce the cardiovascular events and mortality associated with sleep apnea through glycemic and weight control. From a mechanistic perspective, further investigation of the interplay between inflammation, redox pathways, and AMPK signaling will help to establish the main drivers of SF-mediated metabolic dysfunction. 

## Figures and Tables

**Figure 1 cells-13-00470-f001:**
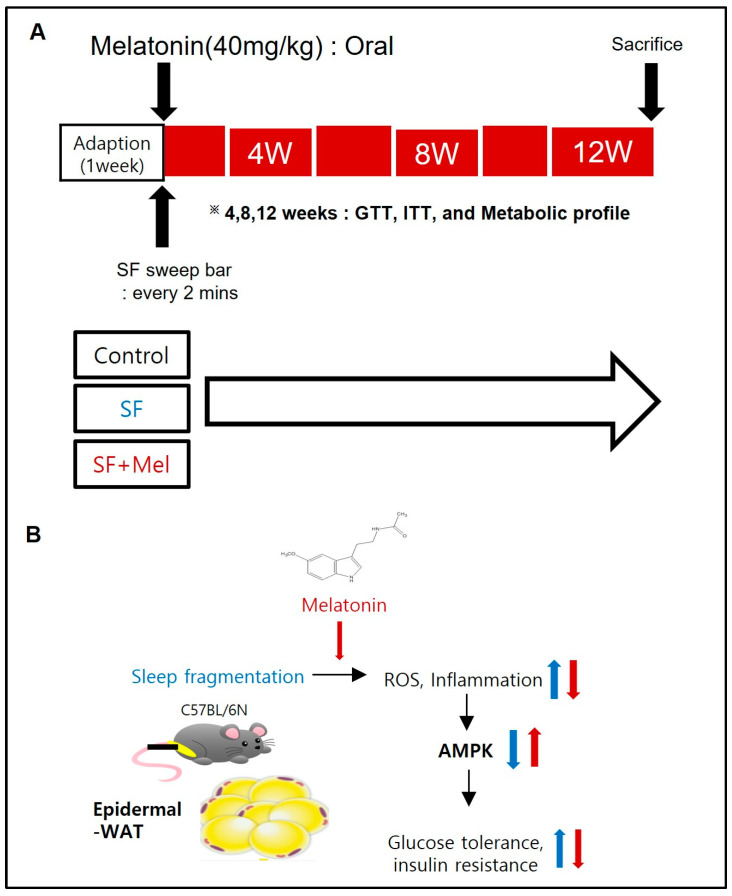
Schematic representation of the experimental designs and graphical summary of the study. (**A**) Schematic of the experimental protocols. SF+Mel mice were housed in the SF chamber, and melatonin was administered by oral gavage at a concentration of 40 mg/kg daily. Subsequently, mice in the control group were housed under standard housing conditions. A glucose tolerance test (GTT) and insulin tolerance test (ITT) were performed at 4 weeks, 8 weeks, and 12 weeks of SF exposure in three groups. (**B**) Proposed design to study the effect of melatonin on SF-induced metabolic changes.

**Figure 2 cells-13-00470-f002:**
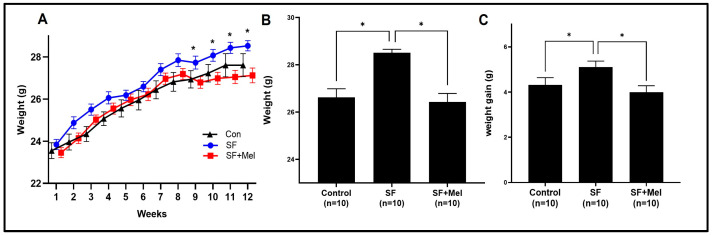
The effects of SF exposure and melatonin treatment on body weight and weight gain were studied in three groups, control, SF, and SF+Mel, for a period of 12 weeks. (**A**) Changes in body weight in control, SF, and SF+Mel groups of mice for 12 weeks. (**B**,**C**) Comparison of body weight and weight gain in three groups of mice after 12 weeks. Changes in body weight and weight gain were compared between the groups using the Mann–Whitney U test. * *p* < 0.05.

**Figure 3 cells-13-00470-f003:**
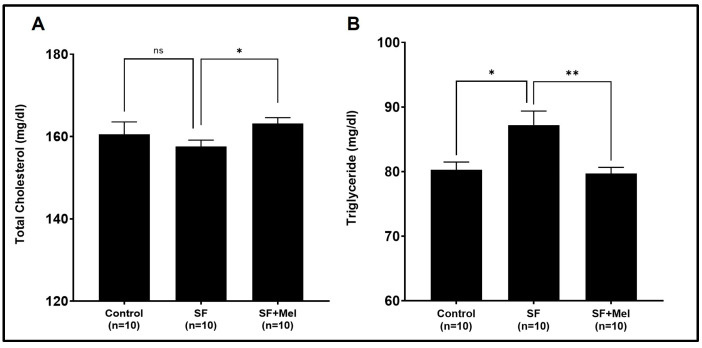
The results of total cholesterol and TG levels after exposure to SF for 12 weeks in three groups. (**A**) Total cholesterol levels in three groups. (**B**) TG levels in three groups. Data are expressed as mean ± SE. Differences were analyzed using Mann–Whitney U-tests. * *p* < 0.05, ** *p* < 0.01, ns; *p* > 0.05.

**Figure 4 cells-13-00470-f004:**
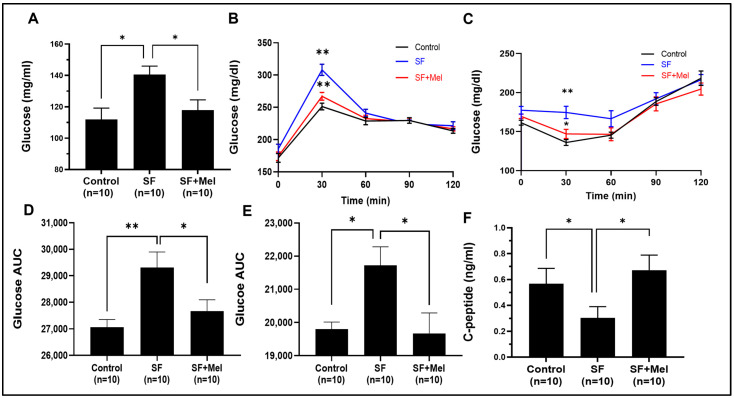
Changes in factors associated with glucose metabolism after exposure to SF and melatonin treatment for 12 weeks in three groups. (**A**) The fasting glucose levels in the control, SF, and SF+Mel group mice for 12 weeks. (**B**) Glucose levels over time according to an oral GTT in three groups of mice. (**C**) Glucose levels over time according to an intraperitoneal ITT in three groups of mice. (**D**,**E**) Glucose AUC derived from GTT and ITT in three groups of mice. (**F**) C-peptide levels in the control, SF, and SF+Mel group mice. The AUC was calculated using the trapezoidal method. Data are expressed as mean ± SE. Mann–Whitney U tests were performed to compare differences. * *p* < 0.05, ** *p* < 0.01.

**Figure 5 cells-13-00470-f005:**
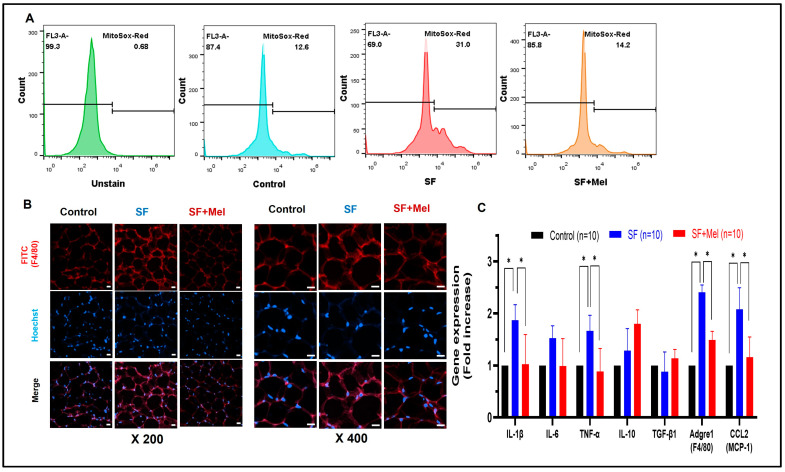
Changes in ROS and inflammation in WAT after exposure to SF and melatonin treatment for 12 weeks in three groups. (**A**) Representative flow cytometry image of ROS in WAT from control, SF, and SF+Mel mice (each sample pooled 3 mice, 3 experiments/each group). (**B**) Representative image of immunofluorescence staining showing the F4/80 positive cells in WAT from control, SF, and SF+Mel mice (5 experiments/each group; nuclei were stained with Hoechst; scale bar: 20 μm). (**C**) Relative gene expression associated with inflammatory pathways using qRT-PCR in WAT of the same experimental groups. Data presented as mean ± SE. Differences were assessed by Mann–Whitney U Tests. * *p* < 0.05. *IL*; interleukin, *TNF*; tumor necrosis factor, *TGF*; transforming growth factor, *Adgre1*; adhesion G protein-coupled receptor E1 (known as F4/80), *CCL2*; chemokine C-C motif ligand 2 (known as *MCP-1*).

**Figure 6 cells-13-00470-f006:**
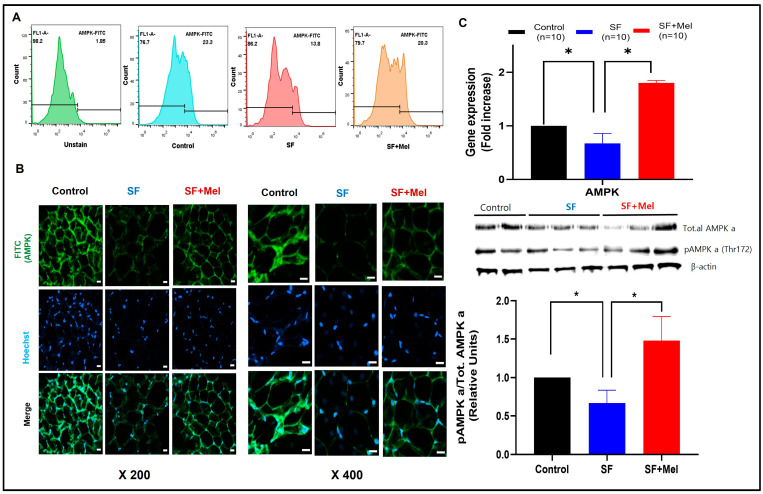
Changes in AMPK levels in WAT after exposure to SF and melatonin treatment for 12 weeks in three groups. (**A**) Representative flow cytometry image of AMPK levels in WAT from control, SF, and SF+Mel mice (each sample pooled 3 mice, with 3 experiments/each group). (**B**) Representative immunofluorescence staining image showing the AMPK levels in WAT from control, SF, and SF+Mel mice (5 experiments/each group; nuclei were stained with Hoechst; scale bar: 20 μm). (**C**) Gene expression and phosphorylation levels of AMPK from WAT in three groups. The gene expression level was measured by RT-qPCR. In addition, the phosphorylation level of AMPK at threonine 172 (pAMPK) was measured by Western blotting (5 experiments/each group). The intensity of the bands was quantified using ImageJ software. Data are expressed as mean ± SE. Differences were analyzed by Mann–Whitney U tests. * *p* < 0.05. AMPK; AMP-activated protein kinase.

**Figure 7 cells-13-00470-f007:**
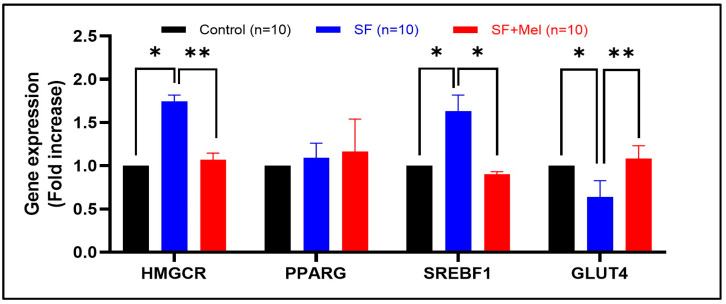
Change in the gene expression levels of glycolipid pathway associated with AMPK in WAT after exposure to SF and melatonin treatment for 12 weeks in three groups. Data are expressed as mean ± SE. Differences were performed using Mann–Whitney U tests. * *p* < 0.05, ** *p* < 0.01. *HMGCR*; 3-hydroxy-3-methylglutaryl-Coenzyme A reductase, *PPARG*; peroxisome proliferator activated receptor gamma, *SREBF1*; sterol regulatory element binding transcription factor 1, *GLUT4*; solute carrier family 2 (facilitated glucose transporter), member 4.

## Data Availability

All data supporting the results are presented in the manuscript. Any other inquiries can be directed to the corresponding authors via email.

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
