# Peer review of "Melatonin Improves Glucose Homeostasis and Insulin Sensitivity by Mitigating Inflammation and Activating AMPK Signaling in a Mouse Model of Sleep Fragmentation"

_cells, 2024, doi:10.3390/cells13060470_

Round 1

Reviewer 1 Report

Comments and Suggestions for Authors

Research has already indicated for some time that sleeping disorders such as obstructive sleep apnea are associated with the development of metabolic syndromes, including development of insulin resistance, glucose intolerance, and type 2 diabetes. While activation of oxidative stress and increased inflammation have been proposed to play a role in the association of sleep disorders and metabolic dysfunction, the underlying mechanism are not yet fully understood. 

The tryptophan-derived hormone melatonin is well known as an important regulator of sleep-wake cycles and as such is used for treatment of some sleep disorders and to reduce jet lag. Melatonin was also reported as a potent antioxidant and free radical scavenger. There is increasing evidence that melatonin interacts with the immune system and has some immunomodulary activity. In addition, a number of studies demonstrated that melatonin has beneficial effects by activating AMP-activated protein kinase (AMPK). 

The significance of the study by Hong et al. is that the well-known effects of melatonin on glucose metabolism, ROS formation, adipose tissue inflammation, and AMPK activation were addressed in a mouse model of sleep fragmentation. 

The experiments appear to be thoroughly performed and technically sound, and the entire study scientifically accurate. The results are well described in the manuscript and the data interpretation is without overstatements. 

In this respect the study is novel, although it cannot provide novel insides into the mechanisms of melatonin activity. For example, for melatonin signaling two G-protein coupled receptors (MT1 and MT2) are known and the respective knock-out mouse models are available and could have been analyzed to address which receptor is essential for the observed effects. In addition, some immunomodulary activity was analyzed. However, there is again no deeper insight into the mechanisms of the immunoregulatory activity of melatonin. For example, the increased presence of F4/80+ cells (as a marker of WAT inflammation) could have been analyzed more precisely for M1/M2 polarity and the cytokine profile towards Th2 or Th17 responses.

However, I consider a very much deeper insight into the mechanisms to be beyond the scope of this study.

Beyond that, there are only a few critical comments and minor recommendations that should be addressed.

Major points of criticism:

1) Number of mice in figure legends

The Material and Methods part states 10 mice for each of the three groups (control, SF, SF+melatonin) and that two separate sets of mice were used to study glycemic response and ROS. It does not become clear how many mice (n=5 or n=10?) are now used in each experiment. Therefore, the legends to the figures should always contain the exact number of mice used in each of the experiments.

2) Statistical Analysis

Most of the time the non-parametric Mann-Whitney U-test was used (for data shown in Figures 2-4, 6, and 7) or the parametric Student’s t-test (Figure 5). However, both tests are only applicable to compare two different conditions. If there are more than two conditions, as here with three, other statistical tests such as ANOVA (for parametric data) or the Kruskal-Wallis test (for non-parametric data) must be used.

3) ROS formation, Figure 5

In Figure 5A only representative FACS plots for ROS formation are shown. Again please specify the number of mice that have been analyzed and combine all measurements in a diagram. The figure could be rearranged, i.e. increase size of Fig. 5C. It is not clear to me why two rather similar panels of immunofluorescence staining with 200 x magnification and 400 x magnification are shown. Does one shows something that can’t be seen in the other? Then it should be highlighted. Otherwise one panel would be enough.

4) AMPK expression, Figure 6

As for ROS formation in Figure 6 only representative FACS plots for the AMPK expression are shown.  Again please specify the number of mice that have been analyzed and combine all FACS measurements in a diagram. Like Figure 5, the Figure 6 should be rearranged and for example Fig. 6C below Fig. 6B and as a right and left panel.

The western blotting should be complemented for the analysis of total AMPK.

Although the AMPK pathway factors were analyzed for gene expression, as shown in Figure 7, it would have been more informative if direct targets of AMP kinase activity had been examined, e.g. by western blotting for phosphorylated substrates such as SREBP1 or ACC.

Minor points that need attention:

5) The reviewer appreciates the self-criticism in the discussion about the limitations of the study. However, the authors also use this skillfully to draw attention to the advantages of the study design.

From here on, however, the discussion gets too out of hand. On the one hand, conclusions are repeated in the penultimate paragraph and in the final conclusion. On the other hand, there is a lot of self-praise. Whether "the study is thorough and well designed,..." and so on, should be left to the judgment of the reviewers. Also the wording “to the best of my knowledge, this is the first study exploring the effect of…” is too obviously fishing for compliments (any published study should contain a novel aspect that has not been published before).

6) Page 4, lines 143ff: Is the staining with Mitosox-red described twice? If this is not by mistake, but intentional, please clarify which staining was used for which assay. 

7) Page 8, line 285ff: The expression of PPARgamma is shown in Figure 7, but is not mentioned in the text.

Comments on the Quality of English Language

There are no major criticisms of the English language and I found very few typos. E.g. page 2, line 54 "Oxidase" should be "oxidase".

Author Response

I am writing to submit the revised version of our manuscript entitled "Melatonin Improves Glucose Homeostasis and Insulin Sensitivity by Mitigating Inflammation and Activating AMPK Signaling in a Mouse Model of Sleep Fragmentation" for consideration for publication in your esteemed journal. We appreciate the valuable feedback provided by the reviewers and have made substantial revisions to address their comments.

Reviewer 2 Report

Comments and Suggestions for Authors

A nice research paper provided beneficial effects of melatonin. The authors provided enough experimental evidences to prove their hypothesis and the presentation is good. They also mentioned some limitations of their study. 

This study will attract future researchers towards the beneficial effects of melatonin, especially sleeping on health, decreasing inflammatory diseases etc. 

I would ask to add a graphical abstract to demonstrate the summary of the study in brief. 

Author Response

Dear Reviewer,

I am writing to submit the revised version of our manuscript entitled "Melatonin Improves Glucose Homeostasis and Insulin Sensitivity by Mitigating Inflammation and Activating AMPK Signaling in a Mouse Model of Sleep Fragmentation" for consideration for publication in your esteemed journal. We appreciate the valuable feedback provided by the reviewers and have made substantial revisions to address their comments.

Reviewer 3 Report

Comments and Suggestions for Authors

The study aimed to understand the mechanisms behind metabolic dysfunction caused by sleep fragmentation (SF). The results showed that chronic SF leads to increased inflammation, oxidative stress, and downregulation of AMPK. However, the changes were mitigated through melatonin supplementation. The authors conducted an excellent study and meticulously analyzed the results, highlighting significant findings. There are only a few minor suggestions for improvement that the authors should consider.

Minor comments:

Line # 71:  Why were only adult male mice used?

Line # 72: DBL laboratory – Where is it located?

Line # 91: 40 mg/kg daily. What is the reason for choosing this concentration?

Line # 111: What is O.C.T compound? The term was used for the first time. Expand it.

Line # 112:  What is FITC? The term was used for the first time. Expand it.

Line # 114: PBS The term used for the first time. Expand it.

Line # 115: min – in earlier and later paragraphs, it was used as minutes – maintain uniformity.

Line # 145: min – in earlier and later paragraphs, it was used as minutes – maintain uniformity.

Line # 192: Triglyceride mentioned as TG in the previous and later lines – maintain uniformity.

Line # 261: Were the AMPK levels measured only after 12 weeks?

Author Response

(The authors gave the same response as above.)
